# “You Can’t Replace That Feeling of Connection to Culture and Country”: Aboriginal and Torres Strait Islander Parents’ Experiences of the COVID-19 Pandemic

**DOI:** 10.3390/ijerph192416724

**Published:** 2022-12-13

**Authors:** Michelle Kennedy, Tess Bright, Simon Graham, Christina Heris, Shannon K. Bennetts, Renee Fiolet, Elise Davis, Kimberley A. Jones, Janine Mohamed, Caroline Atkinson, Catherine Chamberlain

**Affiliations:** 1College of Health, Medicine and Wellbeing, The University of Newcastle, Callaghan, Newcastle, NSW 2308, Australia; 2Hunter Medical Research Institute, Rankin Park, Newcastle, NSW 2287, Australia; 3Indigenous Health Equity Unit, University of Melbourne, Parkville, Melbourne, VIC 3000, Australia; 4Department of Infectious Diseases, Peter Doherty Institute for Infection and Immunity, University of Melbourne, Parkville, Melbourne, VIC 3000, Australia; 5National Centre for Aboriginal and Torres Strait Islander Wellbeing Research, Australian National University, Canberra, ACT 2601, Australia; 6Judith Lumley Centre, School of Nursing and Midwifery, La Trobe University, Bundoora, Melbourne, VIC 3083, Australia; 7Intergenerational Health Group, Murdoch Children’s Research Institute, Parkville, Melbourne, VIC 3052, Australia; 8Lowitja Institute, Melbourne, VIC 3066, Australia; 9We Al-Li Pty Ltd., Goolmangar, Lismore, NSW 2480, Australia; 10Ngangk Yira Institute for Change, Murdoch University, Perth, WA 6150, Australia

**Keywords:** aboriginal health, COVID-19, indigenous health, public health, wellbeing

## Abstract

This Aboriginal-led study explores Aboriginal and Torres Strait Islander parents’ experiences of COVID-19. 110 Aboriginal and Torres Strait Islander parents were interviewed between October 2020 and March 2022. Participants were recruited through community networks and partner health services in South Australia, Victoria, and Northern Territory, Australia. Participants were predominantly female (89%) and based in Victoria (47%) or South Australia (45%). Inductive thematic analysis identified three themes: (1) Changes to daily living; (2) Impact on social and emotional wellbeing; and (3) Disconnection from family, community, and culture. COVID-19 impacted Aboriginal and Torres Strait Islander families. Disruption to cultural practice, and disconnection from country, family, and community was detrimental to wellbeing. These impacts aggravated pre-existing inequalities and may continue to have greater impact on Aboriginal and Torres Strait Islander parents and communities due to intergenerational trauma, stemming from colonisation, violence and dispossession and ongoing systemic racism. We advocate for the development of a framework that ensures an equitable approach to future public health responses for Aboriginal and Torres Strait Islander people.

## 1. Introduction

Aboriginal and Torres Strait Islander people and organisations have been at the forefront of leadership and response from the outset of the Coronavirus (COVID-19) pandemic [1,2]. Aboriginal and Torres Strait Islander Health Services (ATSIHS) had begun developing responses to potential outbreaks prior to the World Health Organization (WHO) declaring COVID-19 a pandemic [3,4], and before the first case had even been reported in Australia in January 2020 [5]. In April of the same year, the United Nations recognised Indigenous peoples globally as a priority population, and highlighted the importance of Indigenous leadership to prevent the spread of COVID-19 [6]. COVID-19 has highlighted the existing health and social inequities stemming from colonisation and to address this, Indigenous people have called for action in response to COVID-19 [2]. This includes recognising and building health policy, practice and research that incorporates equity, culture and Indigenous leadership to develop appropriate emergency responses [7].

In the early stages of the pandemic, Australia and New Zealand had comparatively low numbers of COVID-19 cases and deaths, which have been attributed to early intervention measures such as lockdowns, social distancing, business trading restrictions and international and domestic border closures [8,9]. While clearly effective, these responses pose unique challenges for all parents. Nationwide lockdowns impacted working conditions and unemployment. During the Nationwide lockdown, children were unable to attend school, socialise with others, impacting both child and parental mental health leading to increased rates of parental depression, anxiety and stress during lockdown [10,11]. Rules varied across jurisdiction once the national restrictions eased, with metropolitan Victoria seeing consistent lockdowns, and restrictions that prohibited activities such as attending public playgrounds. Northern Territory and South Australia faced shorter, and less intensive ‘snap lockdowns.’ Despite relatively low infection rates in the Western Pacific region compared to the rest of the world at the time of writing [8], Aboriginal and Torres Strait Islander people are still considered at increased risk of adverse effects from the pandemic, due to intergenerational trauma, stemming from colonisation, violence and dispossession and ongoing systemic racism [12]. Government-implemented control measures have varied between states and territories in Australia since the beginning of the pandemic according to localised outbreaks and political decision making. Some states have implemented strict travel restrictions on tourists and visitors entering remote Aboriginal and Torres Strait Islander communities (see Western Australia [13]). As such, it is likely that Aboriginal and Torres Strait Islander parents and communities have experienced the pandemic differently to other Australians. This has been detailed in a report by the Healing Foundation, which found that survivors of the Stolen Generation are more likely to be adversely impacted by Government restrictions and policies due to past trauma, as well as associations with isolation and decline in health and wellbeing [14].

In support of Aboriginal and Torres Strait Islander leadership in the pandemic, the Australian Partnership for Preparedness Research on Infectious Disease Emergencies Centre of Research Excellence (APPRISE CRE) provided funding for COVID-19 research with First Nations Peoples. This paper is one study with the funded project to Develop a Culturally Responsive Trauma-Informed Public Health Emergency Response Framework for First Nations Communities. This APPRISE-CRE funded project draws on interviews conducted as part of the Healing the Past by Nurturing the Future (HPNF) project, which aims to co-design culturally safe, trauma integrated perinatal care for Aboriginal and Torres Strait Islander parents experiencing complex trauma [15]. In this study, the research team examined Aboriginal and Torres Strait Islander parents’ experiences and the impact of COVID-19 and associated restrictions on them and their children’s lives.

## 2. Materials and Methods

### 2.1. Research Team, Reflexivity and Methodology

This project is led by Aboriginal and Torres Strait Islander researchers and acknowledges that research methodology and research team members’ worldviews influence how the study was conceived, conducted and interpreted. The team consists of Aboriginal researchers, MK (Wiradjuri), SG (Narrunga), CC (Palawa), JM (Narrunga Kaurna), CA (Bundjalung and Yiman), and non-Aboriginal researchers (TB, KJ, ED, RF, CH, SB). The research team embodies extensive experience in qualitative research methods (MK, CH, SB, RF, CC), Aboriginal and Torres Strait Islander health research (MK, CC, SG, JM, CA, CH, RF), Health policy (JM) and parenting and development (CC, RF, SB).

### 2.2. Design, Sample and Recruitment

Ensuring emotional and cultural safety of participants was a key priority [16] with detailed safety protocols. Data were collected over the phone between October 2020 and March 2022 across three Australian states and territories: Northern Territory (NT), South Australia (SA), and Victoria (Vic). These locations were chosen based on existing research relationships. To be eligible for inclusion, participants were required to be: (i) over 16 years of age, (ii) self-identify as Aboriginal and/or Torres Strait Islander, (iii) a parent or expectant parent of a child, and (iv) living in NT, SA or Vic. We focused on parents in this study given it was nested within a larger study, Healing the Past by Nurturing the Future. The experiences of parents during the pandemic were deemed highly relevant given the particular challenges the restrictions were thought to have on families.

Participants were recruited through community networks and Healing the Past by Nurturing the Future partner services using convenience sampling. Flyers were displayed in waiting rooms during routine care. Participating sites distributed information regarding the study for parents to take home and contact the researchers if they were interested in participating, with the option for site staff to record parents’ consent to contact them. Additionally, paid social media advertising was used including Facebook and Twitter. All participants provided informed consent. Each participant received a gift card and children’s book, as well as a small gift for themselves.

### 2.3. Data Collection

Phone interviews were conducted using open-ended and Likert-scale response questions, as part of a broader phone survey, conducted over three interviews (preliminary booking, demographics and COVID-19 experiences; Aboriginal and Torres Strait Islander Complex Trauma and Strengths Questions [15]; International Trauma Questionnaire [17]), with a follow-up wellbeing check SMS and phone-call. This paper reports findings from the open-ended responses in the preliminary COVID-19 experiences interview.

Participants were asked the following open-ended questions during their phone interviews with the research team:What effect has the COVID-19 pandemic had for you?Has the COVID-19 pandemic impacted your capacity for parenting and caring for your children? If so, in what ways?Do you feel the COVID-19 pandemic has affected your children? If so, in what ways?What has been the most helpful so far in trying to cope with the stress of the COVID-19 pandemic?Are there any specific supports that are, or would be, most helpful for you and your family during the COVID-19 pandemic?

Participant responses were directly entered in real time by the interviewer into a purpose-built REDCap survey [18]. Responses were not audio-recorded or transcribed verbatim. Participants were offered a choice of interviewer gender, interviewer Aboriginal and/or Torres Strait Islander status, location and time. Interviewers undertook training prior to conducting interviews and received ongoing supervision to foster reflective practice. Several authors were involved in data collection (TB, EB, CC, RF, KJ, ED), enhancing contextual understanding.

### 2.4. Analysis

Demographic details were described using frequencies and percentages for categorical variables, and mean, standard deviation and range for continuous variables. Open-ended responses were exported into NVivo 12 software for coding. To strengthen our research approach we coded the data using iterative steps [19], including: (1) immersion in data collection and initial line by line coding by two Aboriginal authors (MK/CC) then by TB/RF on a final subsample); (2) develop preliminary codes and organize themes (MK/CC/TB/RF); and (3) discussion of preliminary codes and analytic themes with the authorship team, by reviewing supporting quotes and refining the themes. As Braun and Clarke [20,21] note in their latter work, thematic analysis is not one approach, but rather a set of approaches that are sometimes conflicting. As the qualitative data collected were short responses (often one or two sentences rather than that of an hour-long interview), we applied line-by-line coding, but were nonetheless guided by Braun and Clarkes iterative steps detailed above. Given the long period of data collection and contextual changes during this time, identifiers are included to contextualise these quotes (i.e., age, gender, state, year of interview).

## 3. Results

### 3.1. Participants

179 Aboriginal and/or Torres Strait Islander parents expressed interest in taking part in the study. Of those who expressed interest, 45 were no longer contactable after five attempts, 8 were deemed ineligible, and 16 were no longer interested. A total of 110 participants then completed an interview.

Forty-seven percent of the participants were located in Vic, 45% in SA, and 8% in NT. The majority of participants were mothers (89%), with the mean age being 33.9 years (18–72 years). Demographics are further reported in Table 1. Demographic questions were developed as part of a larger study that examines complex trauma amongst Aboriginal and Torres Strait Islander parents. The demographics provide detail for the reader about the participants in the study.

Three major themes and associated subthemes were identified from the thematic analysis:Disconnection from family, community, and culture: Family and Country, Cultural Practices;Changes to daily living; Access to Services, Remote Working and Learning, Coping Strategies and Valued Supports;Impact on social and emotional wellbeing: Impact on Parents, Impact on Children, Strategies to Mitigate Social and Emotional Impact, Coping Strategies and Valued Supports.

These major themes and subthemes with representative quotes are described below.

### 3.2. Theme 1: Disconnection from Family, Community and Culture

One of the most pressing issues identified by parents in this study was the disconnection from family, community and culture due to the pandemic. For those unable to due to restrictions, not being able to support kin, engage in cultural practices or return to country was considered to be detrimental to the social and mental wellbeing of their children and themselves.

#### 3.2.1. Family and Country

Disconnection from family and country was a significant negative impact of COVID-19 for Aboriginal and Torres Strait Islander families in this study.

“And miss that connection to country to ground ourselves. You realise how much of an impact it actually has. […]. We were able to connect with things online, but it wasn’t that same feeling. You can’t replace that feeling to connection to culture and country…” (#21, Female, 22, Victoria, October 2020)

Many parents reported feeling isolated, not being able to see family, or travel home to country during the pandemic. While some families reported using technology to stay connected, it was not considered sufficient to fill the void of disconnection. Parents expressed that culturally, raising children is a collective responsibility and emphasised the importance of support from extended family. This lack of connection was particularly challenging for new mothers who did not have family present during the birth, or at home after. While some parents who experienced fewer restrictions in their state or territory were able to still have family support, many parents and grandparents reported challenges of not being present to support their family and in particular, support with babies born in the pandemic. Mothers of babies born during or shortly prior the pandemic felt that the lack of connection impacted their children’s knowledge and recognition of who their family are. This had significant impact on usual caring practices for Elders, community, and children.

#### 3.2.2. Cultural Practices

Restrictions during the pandemic heavily impacted usual cultural practices and ceremonies such as births and funerals; with many parents describing distress related to the fact that they “can’t go back to country”. Almost half of the participants indicated that they missed out on a funeral due to COVID-19 restrictions or exposures. One participant discussed the way that the interruption of usual birthing practice and not being able to connect with mob impacted ability to take rest and respite from the baby, and women time.

“In relation to connectedness, us mob when someone has a baby, aunty and cousin come over and help with baby for a little while. They can help with respite and rest. Don’t have the women time. First time mum and don’t know everything.” (#9, Female, 32, Victoria, October 2020)

Many participants explained that they were not able to attend funerals of family and Elders. One participant noted that they were unable to attend the funerals of three Elders who passed during restriction periods, which had “horrific” cultural impacts.

“Isolation has been a big affect, my Aunty passed away and I couldn’t go and see her. I could only speak to her by phone. And couldn’t attend her funeral and other funerals because of Covid”. (#114, Female, 63, South Australia June 2021)

### 3.3. Theme 2: Changes to Daily Living

#### 3.3.1. Accessing Services

Parents in this sample reported impacts on their usual experiences of antenatal and perinatal care. Changes included reduced accessibility to healthcare and medical appointments with people required to test for COVID-19 before attending appointments, and in some cases, restrictions on partners being present during appointments which was challenging for some mothers planning for pregnancy and childbirth. Pregnant mothers were not able to take a support person with them, often omitting their partner from much of the birthing planning processes.

“It has been really hard. I had to go to a lot of appointments alone because I couldn’t take anyone with me to help me to understand what they were saying at the hospital.” (#111, Female, 28, Victoria, October 2021)

Some parents indicated that they struggled to perform COVID-19 tests on their children who did not respond well to having anything near their mouth, and as a result were unable to attend care as they could not prove they were not infected. Many parents discussed the importance of their ATSIHS in providing necessary care, however some reported that COVID-19 also significantly impacted their access to these services. Telehealth was thought to relieve some of the challenges posed by COVID-19 and parents expressed gratitude for telehealth enabling them to maintain health care appointments for themselves and their children. However, this was not ideal for all parents, with some preferring face-to-face engagement with health providers. This was particularly important for appointments regarding mental health.

“I would like to see more face-to-face services for counselling. Video calls are really challenging in this house, there is nowhere to speak privately.” (#144, Female, 50, Victoria, October 2021)

#### 3.3.2. Remote Working and Learning

Parents in this sample reported major changes to work, and their child’s schooling with the implementation of remote working and learning due to the pandemic. Parents reported that working from home arrangements and home schooling had both negative and positive impacts on themselves and their children. For some, juggling home schooling and work commitments was considered “a nightmare”, “difficult”, and “an inconvenience”. Home schooling presented challenges for parents, who reported that the dual role of being a parent and a teacher put strain on relationships and that they found it difficult to keep up with online classes. The challenges of home schooling were particularly difficult for children in the final year of pre-school or first year of school (kindergarten). Parents felt their children missed out on educational opportunities, and some mentioned challenges settling into school when it was possible to attend. Activities such as social events organised by childcare centres and schooling online helped families to stay connected and reduce isolation. Parents reported feeling concern about their children being isolated from their peers, and not able to engage in usual activities such as swimming, going to the park or the beach. This was deemed to particularly impact older children.

Some parents found that they needed to rely on ‘screen time’ to ensure that they could meet their work requirements. However, some parents explained that it provided an opportunity to spend more time with their children. The additional time afforded to bond with children and focus on their child’s education was considered a positive.

“The second lockdown was a bit more tricky because we were both working from home. We allocated different times for each other to work and care for our child”. (#135, Male, 34, South Australia, September 2021)

In this way, parents highlighted the value of connection to family and kin for those whose situation afforded it.

#### 3.3.3. Coping Strategies and Valued Supports

Financial support provided by the Federal Government during the early stages of the pandemic was considered extremely helpful by many parents represented in this study. Families reported financial struggles, which were eased due to the available support packages, including to those who were still able to work, but experienced changed conditions due to COVID-19. Parents suggested that free day-care would have been helpful during the pandemic, especially with wider family and kinship not being able to provide regular support with child rearing, coupled with the closures of schools. Several parents reported high levels of both financial and social support offered by their ATSIHS which was highly valued. Access to food banks and food deliveries were an emotional and financial support from their communities.

“Tangentyere land council has been very supportive with housing and food.” (#23, Female, 33, NT, October 2020)

“We have Rumbalara Co Op they had food parcels from Shepparton because all the Melbourne people took our food and toilet paper. So they’ve been really helpful. And checking in- like a phone call or text message. That was really nice.” (#21, Female, 32, Victoria, October 2020)

### 3.4. Impact on Social and Emotional Wellbeing

#### 3.4.1. Impact on Parents

COVID-19 had substantial impacts on the social and emotional wellbeing of parents. Parents described lockdown periods as “mentally damaging” and expressed experiencing a decline in their mental health.

“I think mentally it was just so hard. There was no space between us all. I didn’t get a break.” (#79, Female, 34, Northern Territory, March 2021)

Parents spoke of heightened anxiety, depression, loneliness, and some even had thoughts of suicide and self-harm. Many of these feelings were related to isolation from friends and family. Parents reported feeling that the pandemic had exacerbated existing mental health issues, or underlying trauma.

“Noticed from a mental health perspective—even though kids are older—my mother is stolen generation—lots of things have come up—doubting my parenting a lot even though I’m not a bad parent—hysterical over guilt that I’ve struggled and let the kids down.” (#154, Female, 47, Victoria, November 2021)

This participant emphasizes the connection between intergenerational trauma and the impact of restrictions on their wellbeing, suggesting the pandemic bought up feelings of associated guilt and struggles. Many reported feelings of fear, stress, worry and uncertainty and described living through the pandemic as being “surreal”. One parent mentioned fear of leaving their home to get supplies due to fear of possible infection. Worry for themselves, their parents, and their children was amplified due to isolation and restrictions on family visits. Parents reported choosing to self-isolate, irrespective of the public health directives, to ensure the safety of themselves and their family, particularly those with babies. Having to balance home, work and school life was tiring for some parents. Parents spoke of “not getting a break” and feeling less patient and more frustrated than usual. One parent spoke of being in “survival mode” and parenting from “an empty bucket”.

“Just being locked in was hard. I don’t have a big support network where we are. My mums in Melbourne and locked in alone. I’m one of five in my family, and I’m the rock. It was a trying time.” (#21, Female, 32, Victoria, October 2020)

#### 3.4.2. Impacts on Children

Parents in this study were mindful of their own mental health, as well as their children’s. Anxiety and fear were commonly reported when reflecting on the impact on their children. The perceived level of pandemic-associated impact on children varied from none at all (mostly those with very young children) to extreme. Some parents noted that COVID-19 and associated lockdowns had exacerbated pre-existing anxieties for their children, while others suggested that it created new anxieties and fears in their children.

“The younger one is very anxious and doesn’t want to leave the house.” (#144, Female, 50, Victoria, October 2021)

One parent described how there had been a number of suicides in their children’s friendship groups, and that their children’s mental health had been significantly affected. Other parents reported limited impacts on their children, stating that they were “too young” and “had not known anything else”.

“My daughter not so much. She’s 3 and is a home body and likes being home constantly. She’s quite independent. My son tremendously. School is his safe space from my mental health, my anxiety and OCD and screaming and crying. Sport has been cancelled, and he trains almost everyday. It really bummed him now that he couldn’t play football and basketball. And summer is just hectic. If we’re in lockdown, it will affect him because he’s gone into routine of training and playing. He can now go to youth centre by himself. And not being able to see other people will have a huge impact on him.” (#53, Female, 30, South Australia, November 2020)

#### 3.4.3. Strategies to Mitigate Social and Emotional Impact

Parents reported using a range of strategies to support themselves and their child’s social and emotional wellbeing. Many parents expressed feeling resilient, turning to new hobbies, and setting goals as ways of maintaining their own wellbeing. Parents reported talking with their children about their feelings or concerns, to try to ease their minds. Partners played an important role in supporting pregnant mothers, and in reducing anxieties and stress. Some parents reported that they opted to not watch the news or worry about the pandemic and to focus on family time instead. Other families felt empowered by knowing the facts and keeping informed with quality sources of information, rather than relying on social media.

“Knowing that facts for my area and where my family was. Miscommunication from media at times was a bit stressful. Always checked facts from South Australian government website.” (#28, Female, 28, South Australia, October 2020)

Parents used strategies such as self-isolation, having somebody else go to the shops, and being generally more careful of exposure to mitigate risk and by extension, ease the stress and anxieties related to concern about the health effects of COVID-19.

#### 3.4.4. Coping Strategies and Valued Supports

Social support and connection were reported as the most important coping strategies for parents. Support from family, parents and community was deemed the most helpful during this time. States and territories enforced strict border closures during different stages of the outbreak restricting movement between states and territories. Some states negotiated a ‘border bubble’ that removed these limitations for people living and working in towns along state and territory borders, allowing people in those areas to move between jurisdictions according to risk. Parents in areas with fewer restrictions found procedures such as “the border bubble” helpful, as it allowed freedom of movement to stay connected to family. Those who lived in regions with tighter restrictions used video calls and social media to maintain connections. Technology played an important role in helping parents during the pandemic; however, it was noted that nothing was able to replace the loss of connection when not being able to visit family or country.

“What could have been nice is wellness packs made available for communities—bush medicine, Aboriginal specific supports.” (#16, Female, 28, Victoria, October 2020)

## 4. Discussion

In our study, Aboriginal and Torres Strait Islander parents described experiences during the pandemic, including disrupted connectedness to family, community, and culture, changes to daily living and difficulty accessing services, and impacts on social and emotional wellbeing for parents and children. Inability to visit country and engage in cultural practice was a pressing issue for Aboriginal and Torres Strait Islander parents. The impact of the pandemic was often dependent on the restrictions implemented in their state or territory, and subject to individual situations, such as whether they were allowed to visit kin or country. Parents described strategies that helped mitigate negative impacts, including financial support and childcare; developing hobbies, talking to others and limited media exposure; connecting using technology and the border bubble. Having the opportunity to spend more time with their children, gaining awareness of their children’s educational needs, flexible work conditions, and pandemic payments were considered beneficial by parents.

Culture and connection to family and community has been well documented as a strength and protective factor for good mental health and wellbeing for Aboriginal and Torres Strait Islander people in Australia [22,23,24,25,26]. These protective factors have been compromised during the COVID-19 pandemic with the introduction of social distancing and lockdown measures [14]. A study of young Aboriginal people living in urban areas found that they described culture as being embedded within relationships, whether this is with family, community or with Elders [24]. This study also found that connection to country was not only about the physical access to their Land but also a space that enabled activities, and that for young, soon-to-be parents, being on Land was important to begin the process of naming their future child [24]. These connections and experiences were disrupted throughout the pandemic, as emphasised by parents in our study.

Australian studies have emphasised increased feelings of isolation, fear, anxiety and hardship [27,28] and inability to access family support during the pandemic [27]. Social isolation was a prominent issue discussed by parents in our study, this is in line with studies of the broader Australian population in which more than half of those surveyed reported feeling more lonely since the COVID-19 pandemic, which was associated with higher levels of social anxiety and depression [29]. Impacts of isolation and quarantine can disproportionately affect Aboriginal and Torres Strait Islander people and communities [30], with studies attributing disconnection from country and culture with decreases in physical health, mental health and wellbeing during the pandemic [14].

Nationwide lockdowns have aggravated pre-existing social and financial inequalities in already at-risk groups, hence equity is a critical consideration for public health responses [31]. We found that financial assistance from the Federal Government when available, was considered extremely beneficial to Aboriginal and Torres Strait Islander parents during the early stages of the pandemic. Certain impacts of the COVID-19 pandemic and associated restrictions may be greater for Aboriginal and Torres Strait Islander parents and communities due to intergenerational trauma, stemming from colonisation, violence and dispossession and ongoing systemic racism, [12] which was further noted in our data in regard to ongoing effects of the Stolen Generation with associations of parenting guilt and struggles. A report on the impact of COVID-19 by the Healing Foundation stated that ‘For those Stolen Generations survivors …. there is an association between isolation, disconnection and a decline in physical and mental health, as well as some (re)triggering of trauma from past and present government policies’ [14]. It is therefore important to consider the individual, collective and historical trauma impacts in future public health emergency responses using an overarching philosophy of cultural humility, safety, and responsiveness.

Our study found reported benefits of digital technologies and telehealth support, however, many parents still felt disconnected from cultural practices and in-person community gatherings. Previous research has reported that Aboriginal and Torres Strait Islander people are avid users of technology and particularly social media [32,33,34]. While telehealth is a useful tool to access services during COVID-19, parents in this study expressed a preference for face-to-face appointments, especially those related to mental health; this echoes findings from other Australian studies [35]. Findings from this study support the need for strategies to strengthen and revitalise systems and for well-functioning accessible social systems. ATSIHS’s have provided health care based on core concepts of holistic support, safety, empowerment, connectedness, collaboration, compassion and care for Aboriginal and Torres Strait Islander people with a focus on meeting specific community needs [36]. Our study coincides with evidence of ATSIHS leadership and responses being in line with community needs, such as including providing food hampers, developing health information, testing and vaccination clinics as well as transfers to hospitals [2,37].

Self-determination is a key recommendation in A National COVID-19 Pandemic Issues Paper on Mental Health and Wellbeing for Aboriginal and Torres Strait Islander Peoples [10]. Specifically, it recommends support and funding for Aboriginal and Torres Strait Islander-led organisations for Aboriginal and Torres Strait Islander -leadership, including in mental health research and support. The critical importance of empowered Aboriginal and Torres Strait Islander communities to address social and emotional wellbeing has been reported [30,38]. However, we believe that these are yet to be embedded in public health responses and argue that there is an urgent need for a culturally responsive and safe, trauma informed public health response framework.

Our study provides policy makers, health care providers and community-based workers with evidence of Aboriginal and Torres Strait Islander parents experiences of COVID-19 and areas for consideration to address inequities in public health responses. Preparedness is a key aspect of outbreaks as having established resources, funding, and services ready can significantly improve not only the response to the outbreak but also reduce the negative impact outbreaks can have of people’s mental health and reduce inequity. Based on these findings we advocate for the development of a culturally responsive and safe, trauma informed framework for advising future public health responses built on equity. Our findings could also provide Indigenous parents and organisations globally with evidence to building their own trauma informed framework to mitigate the lingering negative impacts of current and future pandemics of their Indigenous populations.

## 5. Limitations

Our study was based in three Australian states/territories using convenience sampling and may not be representative of all Aboriginal and/or Torres Strait Islander parents. Further, we focused on parents for this paper, and therefore the experiences may not be representative of the entire Aboriginal and/or Torres Strait Islander community. We were most successful at recruiting parents in South Australia and Victoria, compared with remote areas in the Northern Territory, which may have influenced our findings. The timing of data collection is likely to play an important role in parents’ experiences and their interview responses. We collected data over an 18-month period, and therefore experiences were likely influenced by the varying phases of pandemic in the three jurisdictions. Our participation rate was approximately 60% and parents who chose not to participate may have had different experiences with COVID-19.

## 6. Conclusions

COVID-19 has had a significant impact on daily living, social and emotional wellbeing and connections to community and culture for Aboriginal and Torres Strait Islander parents. Certain elements of the pandemic and restrictions may be greater for Aboriginal and Torres Strait Islander parents and communities due to intergenerational trauma, stemming from colonisation, violence and dispossession and ongoing systemic racism. These impacts demonstrate a need for careful consideration in improving public health responses for future emergencies.

## Figures and Tables

**Table 1 ijerph-19-16724-t001:** Demographics Characteristics.

Demographics	N = 110 (%)
State *	
SA	49 (44.6)
VIC	52 (47.3)
NT	9 (8.2)
Gender	
Female	98 (89.1)
Male	11 (10.0)
Prefer not to say	1 (0.9)
Age, years, range	18–72
Age, years, mean (standard deviation)	33.9 (10.2)
Indigenous status	
Aboriginal	106 (96.4)
Torres Strait Islander	1 (0.9)
Both	3 (2.7)
Relationship status **	
Single	28 (25.7)
Partnered, living together	64 (58.7)
Partnered, not living together	11 (10.1)
Separated/Divorced	6 (5.5)
Number of children living with you ***	
0 (pregnant)	10 (9.1)
0 (children left home)	2 (1.8)
1–2	68 (61.8)
>3	27 (24.6)
Prefer not to say	3 (2.7)
Highest level of education	
Some secondary schooling	14 (12.7)
Completed year 12	7 (6.4)
Other post-school education	61 (55.5)
Completed university degree	28 (25.5)

* SA (South Australia); Vic (Victoria); NT (Northern Territory); ** missing data for one participant; *** missing data for three participants.

## Data Availability

No further data is available to ensure participant confidentiality.

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
