# Peer review of "“You Can’t Replace That Feeling of Connection to Culture and Country”: Aboriginal and Torres Strait Islander Parents’ Experiences of the COVID-19 Pandemic"

_ijerph, 2022, doi:10.3390/ijerph192416724_

Round 1
Reviewer 1 Report
Abstract
Second last sentence needs rewording.
Introduction
Description of COVID-19 related stay at home orders: Given the sampling of participants across three states,it would be helpful to include differences by jurisdiction. Eg the closure of playgrounds is mentioned but of the three jurisdictions included this only occured in Victoria. And within Victoria there was significant differences in restrictions between metro/regional.
Would be good to unpack how intergenerational trauma leads to increased risk of adverse effects from the pandemic. Is this from covid or from the associated restrictions? Well described in Discussion but would be good to expand a little here.
Last sentence - this reads as what is being examined is the impact of COVID-19 itself, whereas it seems the research is broader than this and examining the impacts of the pandemic and associated restrictions.
Would be helpful to provide references for standardised measures mentioned.
Analysis
Would be good to have some more detail here. In the abstract it is mentioned that thematic analysis was undertaken. Thematic analysis is not mentioned in the Analysis section and the descriptions of the analysis seems to be describing a grounded theory method. Would be good to clarify/expand. Manual coding is described. What was NVivo used for?/At what point was NVivo used? What is the difference between a descriptive category and a preliminary code and how did the themes that were developed from both interact? The only reference provided is Braun and Clarke who do not advocate for line by line coding in reflexive thematic analysis. Therefore analysis section needs justification and additional references.
Results
Would be good to have more of a narrative around data presented in Table 1. Why was it considered important to ask about relationship status, number of children and level of education. Presumably there was a rationale for asking and recording this data. Would be good to comment on this data and its relevance to the qualitative findings that follow.
Discussion
I am curious about potential for advocacy beyond the call for the development of a framework and beyond consideration of the next public health emergency. I wonder if the authors see there is learnings/implications for policy and/or practice from the current findings for the current public health emergency, ie, ongoing covid pandemic and linger impacts of previous restrictions. I encourage the authors to be a little bolder in integrating findings and their potential utility.
Overall, a delight to read such thought out clearly articulated research.
Reviewer 2 Report
This paper presents important data about the experiences of COVID restrictions among Aboriginal and Torres Strait Islander parents. The data is high quality and well-presented, and I imagine there were many challenges to collecting this data given the ever-changing Covid restrictions, so credit goes to this team of researchers. The paper is well-written and makes an important contribution. I have a few points of feedback to help clarify some details and logic of the study:
· Introduction and the clarity of the research aim – the data presented appears to be about experiences of the restrictions implemented in response to COVID, and not about other aspects of COVID experiences (eg acquiring or getting sick from COVID, getting vaccinated, trying to prevent Covid, etc). Currently, the aim is written in perhaps too general terms: for example, line 76 the aim is written as “parents experiences and the impact of Covid-19 on them and their children’s lives”. Some tweaking would make the aim clearer that the focus of the paper is about living with Covid restrictions, which would then make a better match with the data presented.
· Introduction: Could the authors clarify the logic of focusing the data collection on parents? For example, why the focus on parents and not on the wider Aboriginal community/communities? Relatedly, what is the link between developing a public health emergency response framework for First Nations communities, and the focus on parents? Could the authors provide a bit more information and reasoning here to help the reader.
· Method: I wondered why the data collection focused on three states/territories and not others? A couple of sentences explaining the reasoning would help.
· Results: This is a minor issue - given that the data is not generalizable, can the authors add minor clarification to their data presentation indicating the findings refer to their sample only. For example line 168, just need to add a few words: “for Aboriginal and Torres Strait Islander families in this sample”. Small tweaks like this throughout will increase the precision in the data presentation.
· Results: The theme of not being able to return to country is important and makes sense, but I imagine some participants lived on country (so did not need to travel back?). Did they have the same disconnection from country? What were their experiences with Covid?
· Discussion - Line 384 impacts of restrictions were greater for some Aboriginal people because of intergenerational trauma…no doubt this is the case for some, but I wasn’t clear where exactly this was evident in the data presented. Could the authors bring this analysis forward in the text. Apologies if I missed this.
Reviewer 3 Report
An interesting paper and an evidence base for developing 'the Framework'
Line 31 please delete 'and'
Although I do not dispute the frequently written statement about the impacts of COVID 19 for Aboriginal and Torres Strait Islanders due to intergenerational trauma, stemming from colonisation, violence and dispossession and ongoing systemic racism (appropriately referenced) it would strengthen this statement if you could refer to any data within your findings (eg quotes) as the manuscript states it was 'noted in the data' (line 387).
Line 382 refers to lockdowns having aggravated pre-existing social and financial inequities (appropriately referenced) but I suggest to include in the discussion the study findings (line 253-256) that Federal Government schemes, when available, did assist financially as this would be an important factor in developing a framework
